# Arthropod biodiversity loss from nitrogen deposition is buffered by natural and semi-natural habitats

Shunxiang Fan[1], Tim Newbold[2], Jan C. Axmacher[3], Charlotte L. Outhwaite[2,4], Yi Zou[5], Zhenrong Yu[1], Yunhui Liu [1]*

1 Beijing Key Laboratory of Biodiversity and Organic Farming, College of Resources and Environmental Sciences, China Agricultural University, Beijing, China, 2 Department of Genetics, Centre for Biodiversity and Environment Research, Evolution & Environment, University College London, London, United Kingdom, 3 Department of Geography, University College London, London, United Kingdom, 4 Institute of Zoology, Zoological Society of London, London, United Kingdom, 5 Department of Health and Environmental Sciences, School of Science, Xi'an Jiaotong-Liverpool University, Suzhou, China

* liuyh@cau.edu.cn

## Abstract

Nitrogen (N) deposition is known to strongly modify biogeochemical cycles and trophic interactions, in turn altering ecosystem functioning and plant diversity around the globe. However, our understanding of N deposition effects on arthropod diversity remains limited. Here, we investigate how N deposition impacts the diversity of arthropods by combining biodiversity data from the PREDICTS database with data on global N deposition and land cover using mixed-effects models. We then explore the potential for semi-natural and natural habitats ('SNH') to buffer against potential N deposition-linked biodiversity losses. N deposition has a negative effect on arthropod biodiversity. Both, species richness and abundance are significantly reduced in areas of high levels of N deposition when compared to areas of low N deposition, with responses varying across different land-use types. The strongest negative effects of N deposition on arthropod diversity were observed in locations where the local land use entails the least anthropogenic modification. At the same time, with the exception of cropland-dominated landscapes, increases in the amount of SNH in the surrounding landscape reduced arthropod biodiversity losses associated with N deposition. We conclude that SNH can play an important role in mitigating the negative effects of N deposition on arthropod diversity, with the conservation and creation of these habitats promoting arthropod diversity even under high levels of N deposition.

## Introduction

Nitrogen (N) is an essential nutrient for plant growth and productivity, and its availability strongly affects ecosystem functioning by influencing interspecific competition and trophic relationships [1–3]. N inputs can affect plant primary productivity, species

**Data availability statement:** The data that support the findings of this study are openly available in figshare at DOI: https://doi.org/10.6084/m9.figshare.29109170.

**Funding:** This study was supported by National Natural Science Foundation of China (grant number: U24A20412 and grant number: 41871186) (https://www.nsfc.gov.cn/) (received by YL); China Scholarship Council (grant number: 202306350102) (https://www.csc.edu.cn/) (received by SF) and Natural Environment Research Council (grant number: NE/R010811/1) (https://nerc.ukri.org/) (received by TN). The funder had no role in study design, data collection and analysis, decision to publish, or preparation of the manuscript.

**Competing interests:** The authors have declared that no competing interests exist.

**Abbreviations:** CR, Cropland; MAP, mean annual precipitation; MAT, mean annual temperature; PA, Pasture; PV, Primary vegetation; PF, Plantation forest; SNH, semi-natural and natural habitats; SV, Secondary vegetation; UR, Urban; VIFs, variance inflation factors.

composition, vegetation structure, litter load, duff characteristics, fire regimes, and soil properties [4–7], which can further reshape biotic communities, impacting micro-fauna, flora, and fauna globally. Naturally, atmospheric N deposition ($N_{dep}$) [8] and microbial N fixation [1] form key N sources in terrestrial ecosystems. Human activities, notably the burning of fossil fuels and chemical fertilizer applications, have at least doubled total average N deposition (Human-induced N input) rates in less than a century [9]. Although current N deposition has been proposed as a major threat to global terrestrial biodiversity [10–12] and has also been identified as one of the five major drivers hindering the achievement of global biodiversity conservation targets [13], its global-scale effects have remained underexplored [14], especially when compared to the increasingly well-recognized impacts land use and climate change have on biodiversity [15–19]. Global N deposition levels continue to increase, despite ongoing efforts to reduce them [20–22], implying that any associated potential threats to biodiversity are likely to increase. A better understanding of the nature and strength of N deposition impacts on the diversity especially of arthropods, the world's most species-rich taxa, are therefore critical to inform effective global biodiversity conservation planning.

Previous studies on the impacts of N deposition on biodiversity have mostly focused on plant species and local scales. Relying strongly on data from field N deposition experiments, they show that high N deposition rates trigger losses in plant diversity and changes in vegetation composition [23–28]. While research into responses of animal assemblages to N deposition have remained limited [29–31], it might be expected that plant community responses translate up to higher trophic levels.

N deposition rates are furthermore impacted by land use and landscape patterns. For example, the conversion of agricultural land to semi-natural forest or grassland ecosystems often reduces regional N deposition rates on remaining agricultural land due to changing ground surface features (surface roughness and soil moisture levels), which are important for dry and wet N deposition [32]. Nonetheless, large-scale responses of animal biodiversity to N deposition under different land use types remain poorly understood. Understanding these responses is further complicated by temporal variations in the relative proportions of reduced and oxidized N deposition [33] which can result in significant variations in the ecological effects linked to overall N depositions [34]. A better understanding of specific ecological impacts of these two forms of N deposition can support a more targeted and effective biodiversity conservation.

Arthropods represent the most species-rich taxa of terrestrial animals and play critical roles in ecosystem functions, ranging from pollination, decomposition and nutrient cycling, pest control, soil aeration, and fertility to seed dispersal [35]. A growing body of literature indicates a continued decline in their diversity [5,14,18,36,37] with cascading effects on ecosystems, and ultimately, human well-being [35]. To date, the impact of N addition on arthropods has only been studied at highly localized scales. In this context, elevated N levels have been indicated to heighten risks of pest outbreaks through enhanced foliar palatability [38], while negatively impacting

pollinating insects by altering the chemical makeup of nectar and pollen [39,40], indirectly resulting in a decline in pollination services and crop yields [41]. On a landscape scale, atmospheric N deposition may increase particularly in cropland (CR)-dominated landscapes, with crop management linked to key additional sources of N deposition [42] that will affect biodiversity also in the surrounding areas [43]. Understanding N deposition impacts on arthropod communities on a global scale can in this context provide critical new insights for safeguarding associated ecosystem services particularly within agricultural landscapes.

In agricultural landscapes, semi-natural and natural habitats (hereafter SNH) are important for the maintenance of biodiversity and associated ecosystem services [44–47]. Their presence is already known to mitigate adverse effects of increasing temperature [18,46] and intensive agricultural production [48,49] on arthropod assemblages. SNH are believed to provide general refuge spaces for species under pressure from intensive cultivation practices, including from high N fertilizer applications on adjacent agricultural land. Therefore, SNH are expected to buffer detrimental effects of N deposition on agro-biodiversity. However, the expansion of agricultural land in response to increases in the human population and changing consumption patterns has led to an ongoing loss of SNH and increases in CR areas that also result in increased local N deposition rates [50]. Such changes may disproportionately contribute to a sustained decline in species diversity [50,51], thereby posing a threat to plant and animal taxa, and challenging our ability to reach global biodiversity conservation goals [52]. It is therefore essential to explore if and how SNHs buffer the negative effects of N deposition (from now on called "buffer effect").

To explore this question, we combined data on global N deposition and the composition of arthropod assemblages across different land-use types to assess the relationship between N deposition and biodiversity (measured as species richness, with total abundance of arthropod assemblages also considered). Specifically, we extracted arthropod diversity data for 6,416 sites (including site-level land use information) from the PREDICTS database (Fig 1). Using the coordinates of each site, we then extracted data on N deposition, proportion of SNH and CR, and other covariates (temperature, precipitation, soil pH, N fertilizer). Based on this information, we conducted analyses using mixed-effects models to answer two core sets of questions: (1) How does N deposition impact arthropod species richness and total abundance across

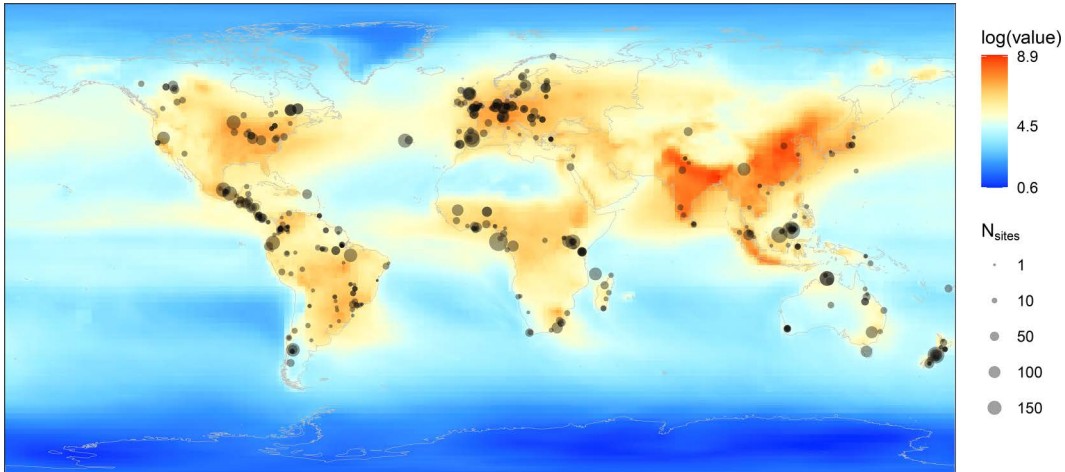

**Fig 1. The distribution of sites from the PREDICTS database and the level of total N deposition.** The size of the point represents the number of sites per study ($N_{sites}$). The change from blue to orange indicates the increase in the total amount of atmospheric N deposition. N deposition values were $\log_e$-transformed, with N deposition ranging from 0.8 mg N m$^{-2}$ yr$^{-1}$ to 7960.0 mg N m$^{-2}$ yr$^{-1}$. The outline map is based on Natural Earth Data (http://www.naturalearthdata.com/about/terms-of-use/), which is published under a CC-BY 4.0 license. The data underlying this figure can be found in https://doi.org/10.6084/m9.figshare.29109170.

different land-use types? In this context, does the deposition of reduced N (ammonia) exert a greater influence on arthropods than that of oxidized N (e.g., nitrous oxide)? (2) Can the presence and proportion of SNH in the surrounding landscape buffer detrimental effects of N deposition on arthropod diversity, particularly within CR sites? We hypothesize that (1) With the increase in N deposition, arthropod diversity generally declines irrespective of land-use type. When comparing different forms of N deposition, reduced N deposition exhibits more significant adverse effects on biodiversity relative to oxidized N deposition, primarily due to its pronounced ability to induce soil acidification. (2) SNH can buffer against the negative effect N deposition exerts on arthropod diversity.

## Methods

### Arthropod biodiversity and land use data

Data on the composition of arthropod assemblages across different land-use types were extracted from the PREDICTS database [53,54]. This global terrestrial biodiversity database contains spatial comparisons of local biodiversity from sites of differing land use and/or land-use intensity. The complete dataset includes 480 data sources (published articles, or unpublished datasets, but with published methodology) with ~3.2 million species records, covering all major land regions of the world except Antarctica. The database covers the years 1984–2013, with most data concentrated after 2000 [55]. It has a hierarchical structure, with each data source containing one or more 'studies', where data were collected using the same method (666 studies in total), each study containing one or more spatial 'blocks', representing distinct spatial clusters of sites, and each block containing a number of specific sampling 'sites'. Each site has records of species abundance or occurrence, from which it is possible to derive community composition, and each study site is also assigned a standardized land-use type (see below for details) [53,55,56]. Almost all sampled sites within PREDICTS are accompanied by geographical coordinates giving their precise location.

In our study, we selected data for Arthropod species in the PREDICTS database, including six taxonomic classes: Arachnida, Chilopoda, Diplopoda, Entognatha, Insecta, and Malacostraca, which together encompassed 1,121,105 records. We calculated two site-level measures to use as response variables in our models: total abundance (total number of individuals across all species sampled at a site) and species richness (the total number of species—as defined by the original study authors—sampled at a site). Abundance metrics are highly dependent on sampling effort and require standardization to enable meaningful comparisons. In contrast, the relationship between species richness and sampling effort is non-linear and prone to a saturation effect. If the sampling effort varied across sites within a study, sampling effort was scaled so that the highest value of sampling effort was 1 and all others were relative to that. The abundance measurements were then divided by this relative sampling effort. We therefore assume that there is a linear relationship between recorded abundance and sampling effort (following Newbold and colleagues [57]—it should nonetheless be noted that, for very large samples, this pattern is likely asymptotic once approaching a state where all specimens occurring in a locality have been sampled). Sites with the same coordinates and land-use conditions were merged to combine sub-samples from the same location (e.g., multiple pitfall traps in an array of traps). As sample sizes are incomplete and vary, we also calculated the Chao-1 estimator [58]. While being aware of the limitations of species estimators in comparing incompletely sampled communities, we employed the Chao-1 estimator as a surrogate to verify the consistency of our results with the observed species richness. The patterns of the results were broadly similar to those presented in the main text (the results of the models based on Chao-estimated richness are presented in S3 Text).

The PREDICTS database assigns sites to specific land-use classes based on information provided in the source publication or by enquiry to the original study authors. Land uses are divided into six categories: (1) Primary vegetation (PV), comprising forest or non-forested natural habitat that is not known to have been destroyed in the past; (2) Secondary vegetation (SV), which represents natural habitats that have been destroyed by human activities or extreme natural events in the past, but where vegetation is being restored or naturally recovering; (3) CR, representing land used to grow

herbaceous crops; (4) Plantation forest (PF), which represents areas used to grow timber or crop trees or shrubs for commercial or subsistence use; (5) Pasture (PA), comprising land known to be regularly or permanently grazed by livestock, and (6) Urban (UR), comprising areas covered with human habitation, buildings, or areas managed for recreation (e.g., parks). To study the effect of SNH on arthropod diversity responses to N deposition in the surrounding area, we focus on areas that are not dominated by impervious surfaces, thus excluding UR areas. We also excluded records with undetermined land use, and records missing either geographic coordinates or abundance data. The final dataset included total abundance and species richness data of 6,416 sampled sites, representing 270 studies.

## Nitrogen deposition datasets

We selected and used outputs of the following 11 global deposition models from the second-phase mission of the Hemispheric Transport of Air Pollution (HTAP II) project [59,60] to obtain estimates of N deposition, following the criteria of Tan and colleagues [60]: CAM-Chem, C-IFS_v2, SPRINTARS, GOCARTv5, OsloCTM3v.2, GEOSCHEMAJOINT, GEOS5, EMEP_rv48, GEM-MACH, CHASER_re1, and CHASER_t106 re1 (Details in Tan and colleagues [60]; Schwede and colleagues [34]). These 11 models were selected as they satisfied the following two criteria suggested by Tan and colleagues [60]: (1) The modeled range of N deposition amounts for the 11 models was limited to ±20% of the maximum and minimum measured N emission values, respectively (emission inventory grids are provided for seven different sectors, e.g., energy, industry); (2) Compared with the mean of all modeling deposition values, the modeled values of the these 11 models did not exceed ±1.5 times the interquartile range of the multi-model average values (see further details in Tan and colleagues [60]). These outputs of N deposition were interpolated into a 1° × 1° grid (resolution of ~111 km at the equator) by taking the arithmetic mean of values from these 11 models (Fig 1). The dataset shows strong consistency with N deposition data from other datasets and observation networks [34,42,60]. The dataset provides two types of N deposition data: oxidized nitrogen forms ($NO_y$, primarily generated from high-temperature processes including industrial processes and fossil-fuel combustion), and reduced nitrogen forms ($NH_x$, primarily originating from agricultural fertilizers and animal manure). The total N deposition was obtained by summing the estimates of oxidized N deposition and of reduced N deposition. Values of total oxidized and reduced N deposition for each sampling site were then extracted based on geographic coordinates. In addition to total N deposition, we compared the responses of arthropod diversity to the two main types of N deposition, separately (results are presented in S2 Fig, S5–S8 Tables). Although data with a higher spatial resolution would be helpful for more precise results when relating field-sampled arthropod diversity to N deposition, global estimates of N deposition are not currently available at a spatial resolution finer than 1°. We checked the consistency of our results by running models for Europe, where 0.1° × 0.1° estimates of N deposition are available, and found that the results using the fine-resolution N deposition data were consistent with those using the coarse-resolution data (see S2 Text for more details).

## Other variables potentially associated with N deposition impacts

In order to enhance the model's explanatory ability, we also considered additional factors (climate ~temperature and precipitation, soil pH, N fertilizer application) that may potentially influence the interaction between land use and N deposition in our study [26,42,61–63]. For climate data, we extracted estimates of mean annual temperature (MAT) and mean annual precipitation (MAP) for 2010 from CRU TS version 4.07 [64] at 0.5° spatial resolution. To maintain consistency with the year of N deposition data, the year 2010 was also chosen for MAT and MAP data. The CRU TS dataset is a global meteorological dataset for all land regions except Antarctica, derived from interpolated monthly climate anomalies based on extensive networks of weather-station observations. It covers the period from 1901 to the present day, with an annual timestep. To obtain data on N fertilizer application, we utilized the Global Fertilizer and Manure database, Version 1, which combines national levels data of crop fertilizer use obtained from the International Fertilizer Industry Association (IFA) with

global harvested areas of 175 crops at a resolution of 0.5 degrees [65]. Notably, N deposition data are closely linked to, and partially derived from, agricultural N fertilizer application [42,66]. Therefore, the spatial patterns of N deposition and fertilizer application are highly correlated [67] and can also be used as general intensity indicators of anthropogenic influence on ecosystems. We compared N fertilizer data with N deposition data in model selection to assess their significant effects on arthropod diversity in this study. Soil pH data was extracted from SoilGrids250m version 2.0 [68]. It models the spatial distribution of global soil properties using global soil profile information and covariate data by machine learning to generate a collection of world soil attribute maps with a resolution of 250 m. The four variables (MAT, MAP, soil pH, and N fertilizer) were extracted based on the geographic coordinates of each sample site.

## Proportion of SNH and cropland in the surrounding landscape

We used global fine-scale estimates of land use for 2005 [69] to extract the proportion of SNH surrounding each sampling site. We selected the 2005 data as it was closest to the median sampling year of the PREDICTS sites included in our analysis, and the land-use categories are also consistent with those recorded in the PREDICTS database. A similar approach has been used in other PREDICTS-related studies [18,70]. For the land-use dataset, we divided the terrestrial area into five types, corresponding with the classification used in PREDICTS—PV, SV, PA, CR and UR. The dataset was created by statistically downscaling mapped coarse-scale land-use data (the 0.5° Land-use Harmonization data [71]) based on their relationships with climate, topography, soil, land cover, population density, and accessibility. The downscaled dataset provides global mapped estimates of land use at a 30-arc-min spatial resolution (approximately 1 km × 1 km at the equator) [69]. PV and SV were grouped together as SNH. To estimate the proportion of SNH in the landscape surrounding each study site, we created a 5-km circular buffer around the midpoint of each sampling location and calculated the mean percentage of PV and SV within the buffer. The proportion of CR was also extracted using the same method and dataset.

## Statistical analysis

We used mixed-effects models to explore the impact of N deposition on arthropod total abundance and species richness. N deposition (total) was treated as a fixed effects, either combined or separated by oxidized and reduced forms. Continuous variables, including N deposition, MAT, MAP, soil pH, N fertilizer application, percentage of SNH, and percentage of CR in the surrounding landscape were standardized using z-transformation to convert them into standardized, dimensionless values, and they were also included as part of fixed effects. Study identity, and spatial block nested within study, were included as random effects to account for the differences in sampling methods used among studies, and the spatial distribution of sites within studies, respectively [17]. The total abundance data contained many non-integer values, and thus we used a natural logarithm transformation prior to fitting the data into a linear mixed-effects model with normally distributed errors. For species richness, generalized linear mixed-effects models with Poisson-distributed errors were used. Additionally, we also incorporated a random intercept of site identity in species richness model to control for overdispersion [72]. For more detailed information on the models' output, refer to S3–S4 Tables. To illustrate the relative magnitudes of arthropod diversity responses associated with increasing N deposition, we calculated the model-estimated percentage difference in both total abundance and species richness across the range of N deposition values at the sites included in the analyses. We set the 2.5th percentile value of total N depositions as the reference value, i.e., as the value representing the low N deposition in this study, and sampled 10,000 estimates based on the model variance–covariance matrix to predict differences in abundance and richness compared to this reference. Median values and 95% confidence intervals thus represent the percentage difference compared to this baseline condition. Since results were similar for the separate models of oxidized and reduced N deposition (see S5–S8 Tables), we only used the total N deposition for subsequent analyses investigating its interactive effect with land use, proportion of SNH and proportion of CR (hereafter, N deposition therefore refers to total N deposition).

To address our hypotheses, we built the minimum adequate mixed-effect model associated with N deposition for species richness and total abundance, respectively, by backward stepwise selection. Initially, we tested the interaction terms based on the most complex fixed-effects structure model. Subsequently, we removed the interactive effects before examining the main effects. If the interaction effects are significant, the main effects will be retained regardless of their importance (S1 Text). Specifically, we included N deposition as a fixed effect to explore its effect on arthropod species richness and total abundance. N fertilizer was also included as a fixed effect in model selection to test whether it significantly influences arthropod diversity. Besides, other factors and their interaction with N deposition, including MAT, MAP, and soil pH, which have been reported to potentially influence arthropod diversity as well [42,61–63], were included as covariables in the model. Additionally, to further investigate variations in the impact of N deposition across different land-use types (In our study, land use was a 5-levels categorical variable ~PV, SV, CR, PF, and PA), we incorporated N deposition and its interaction with each land-use type into our analysis. Further, N deposition and percentage of SNH, as well as the proportion of CR were also included as interacting fixed effects in the model, respectively, to investigate potential interactions between N deposition and SNH/CR within a 5-km radius around each site on arthropod species richness and total abundance. Finally, we also added a three-way interaction between N deposition, the proportion of SNH, and land use, to further explore whether the buffering effect of the proportion of SNH varied across land use. While it would be valuable to investigate how the buffering effects of SNH are influenced by climatic or soil factors, incorporating such variables would introduce higher-order interactions that could render the model excessively complex and challenging to interpret. Given that this was not the primary focus of our study, we did not to include these interactions in the model. Similarly, due to the potential for increased complexity, we did not incorporate different N deposition models as a factor. Model formulae and model selection can be found in S1 Text. We used variance inflation factors (VIFs) to assess collinearity among predictor variables. VIFs were all lower than 10, indicating low collinearity between predictors [73] (S12 and S13 Tables).

We checked all the fitted models to assess the extent to which they met the model assumptions. All models were validated by checking their residual distribution following suggestions from Zuur and colleagues [73], and we used Moran's I to test for study-level spatial autocorrelation in the residuals (S4 and S5 Figs). All statistical analyses were conducted in R 4.03 [74], with the models run using the lme4 package version 1.1-30 [75], the extraction of continuous raster data using the raster package version 3.5-15 [76], and the predictsFunctions package version 1.0 for final model selection and PREDICTS data processing [77].

### Data sources

Datasets used in our study: The PREDICTS data can be downloaded at https://Data.nhm.ac.uk/dataset/The-2016-release-of-The-PREDICTS-database. The Nitrogen deposition data can be downloaded at https://thredds.met.no/thredds/catalog/data/EMEP/Articles_data/Schwede_etal_Ndep_2018/catalog.html. The land-use dataset used in our study to extract the proportion of SNH and CR in surrounding landscape can be downloaded at https://doi.org/10.4225/08/56D-CD9249B224. The climate data (MAT and MAP) were extracted from CRU TS version 4.07 at https://crudata.uea.ac.uk/cru/data/hrg/index.htm. The nitrogen fertilizer application data (Global Fertilizer and Manure, Version 1) can be downloaded at https://sedac.ciesin.columbia.edu/data/set/ferman-v1-nitrogen-fertilizer-application/data-download. The soil pH (SoilGrids250m version 2.0) data can be downloaded at https://soilgrids.org/.

### Results

#### Land use-specific effects of N deposition on arthropod diversity

Across all land uses, arthropod biodiversity showed a significant decline with increasing N deposition both in terms of species richness and total abundance (Fig 2). N deposition exhibited a significant negative effect on arthropod species richness in PV ($p = 0.009$; S3 Table). Specifically, species richness at high-N deposition sites was 39% lower than at low-N deposition sites within PV (Fig 2A). Species richness was about 24% lower in CR sites experiencing high total

atmospheric N deposition (2326.3 mg N m$^{-2}$ yr$^{-1}$, representing the 97.5th percentile value of total N deposition in CR) than at sites experiencing low N deposition (as the 2.5th percentile value of total N deposition) (Fig 2A). Species richness also shows significant declines in SV (23%, $p < 0.001$) and PA (20%, $p = 0.011$), with both ranking just below CR in terms of impact. Negative responses of arthropod species richness to N deposition were observed to be weakest in PF (only decline 0.7%, $p < 0.001$, Fig 2A and S3 Table). Total abundance also showed varied responses to N deposition among the different land-use types (Fig 2B and S4 Table), with the greatest losses observed in PA (decline 65%, $p < 0.001$, S4 Table), followed by PV and then SV (Fig 2B and S4 Table). The effect of N deposition on total abundance in sites of CR (decline 34%, $p = 0.024$, S4 Table) and PF (decline 38%, $p < 0.001$, S4 Table) were significantly lower compared to that in sites classed as the other three land-use types.

Similar responses of arthropod diversity to land use and N deposition were also observed when reduced N deposition and oxidized N deposition were analyzed separately (S5-S8 Tables). However, reduced N deposition had a stronger impact on species diversity (for species richness model $x^2_{22,23} = 23.13$, $p < 0.001$; total abundance model $x^2_{21,22} = 16.71$, $p < 0.001$), while impacts from oxidized N deposition were weaker (for species richness model $x^2_{22,23} = 8.04$, $p = 0.004$; total abundance model $x^2_{21,22} = 10.10$, $p = 0.001$). For example, when evaluating the impacts of N deposition on arthropod species richness in CR, reduced N deposition leads to a 12% greater decline compared to oxidized N deposition.

### The effect of SNH in buffering the impact of N deposition on arthropod diversity

Increases of N deposition had a less pronounced negative effect on species richness as SNH coverage in the surrounding landscape increased (Fig 3). These SNH proportion-linked buffering effects were observed for all land-use types except CR, while the strength of associated arthropod diversity benefits varied among land-use types (Fig 3A). Buffering effects were stronger in PV and SV than in PA and Planation forest (Fig 3Aa, 3Ab, 3Ad, and 3Ae). Although species richness in CR generally benefits from a high percentage of SNH, no buffering effect against N deposition was observed, as the declining trend remained unaffected by increased SNH (Fig 3Ac). For total abundance, the buffering effects of increased

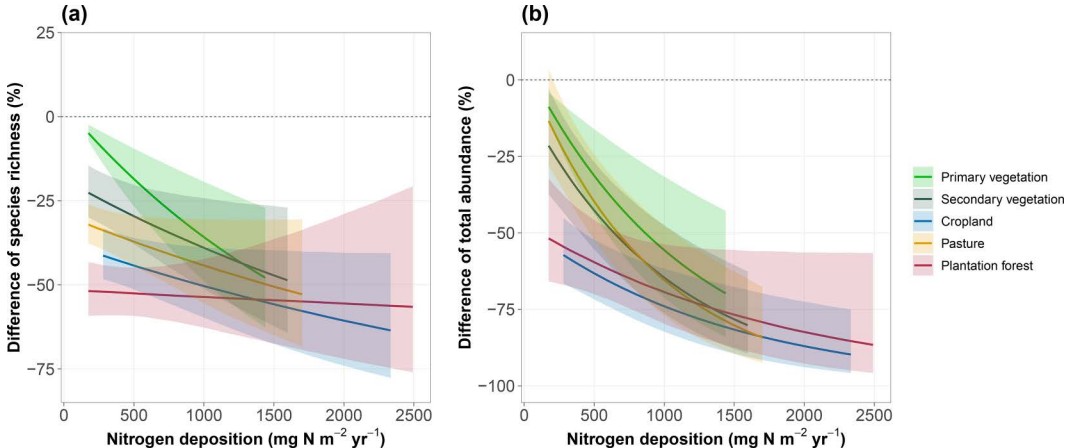

**Fig 2. The response of arthropod species richness (a) and total abundance (b) to the interaction of N deposition and land use.** Likelihood ratio test of the interactive effect of N deposition and land use: species richness $x^2_{19,23} = 35.62$, $p < 0.001$; total abundance $x^2_{18,22} = 30.68$, $p < 0.001$. Values represent the percentage difference compared with Primary vegetation sites with the 2.5th percentile value of total N deposition. The lines with different colors represent the median predicted value for each land-use type, with shaded areas representing the 95% confidence intervals. The results are predicted across 95% of the range of N deposition values for each land-use type. The number of sites were: Primary vegetation, $N_{site} = 1,615$; Secondary vegetation, $N_{site} = 1,511$; Cropland, $N_{site} = 1,563$; Pasture, $N_{site} = 1,347$; Plantation forest, $N_{site} = 380$. The N deposition range used for plotting covers from 2.5% to 97.5% of sampled sites for all land-use types included in the model: 109.4–2491.6 mg N m$^{-2}$ yr$^{-1}$. The data underlying this figure can be found in https://doi.org/10.6084/m9.figshare.29109170.

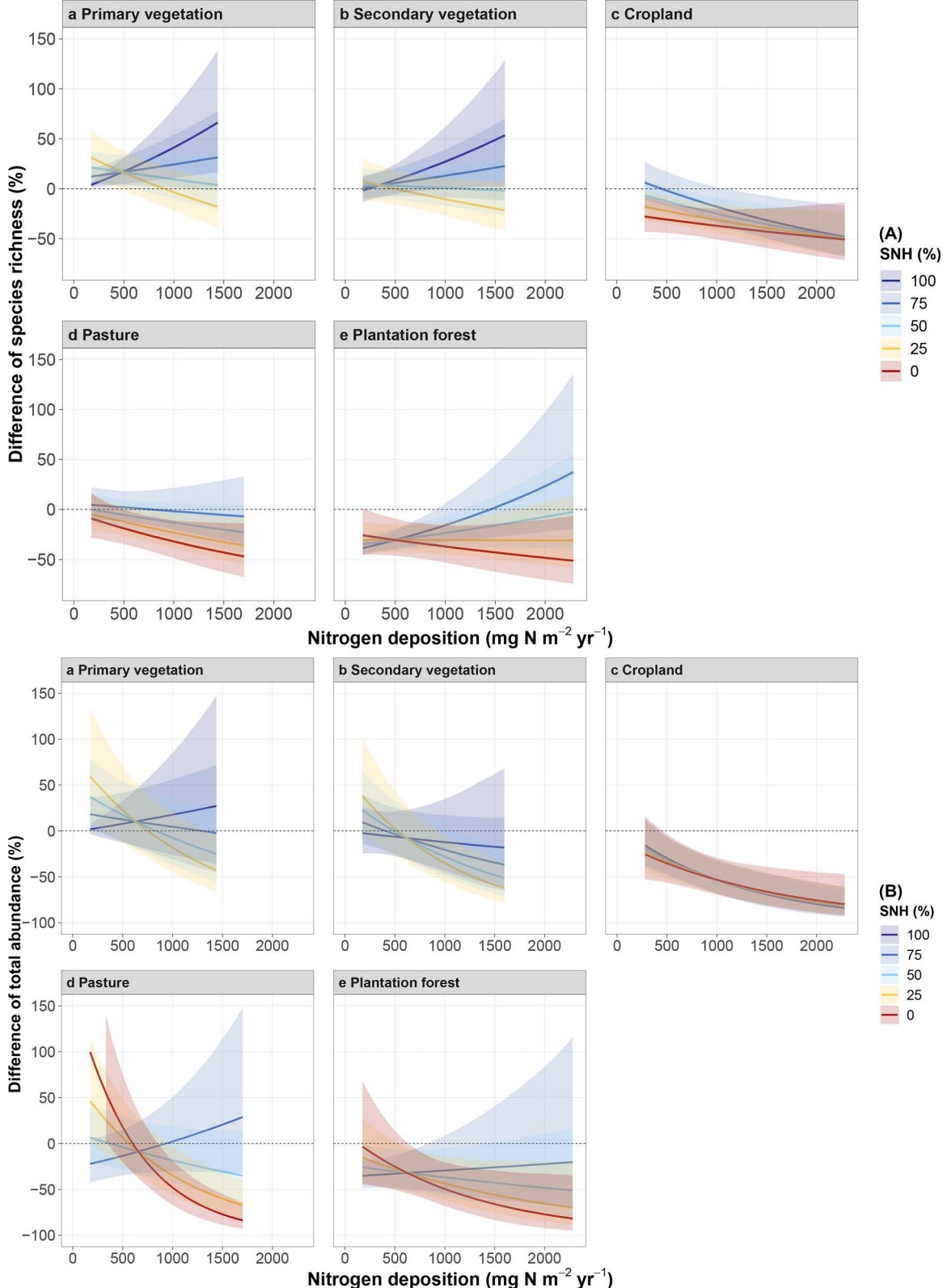

**Fig 3. The interactive effect of SNH and N deposition on arthropod species richness (A) and total abundance (B) in different land-use types.** Likelihood ratio test of the interactive effect of N deposition and proportion of SNH in different land use types: species richness $x^2_{18,23} = 40.09$,

$p = p < 0.001$; total abundance $x^2_{17,22} = 33.81$, $p < 0.001$. Values represent the percentage difference compared with Primary vegetation with the 2.5th percentile value of total N deposition and 100% surrounding SNH. The lines with different colors represent the median predicted value of different percentages of SNH in the surrounding landscape, with shaded areas representing the 95% confidence intervals. The data underlying this figure can be found in https://doi.org/10.6084/m9.figshare.29109170.

SNH were also observed in sites of PA, PF, SV, and PV (Fig 3Ba, 3Bb, 3Bd, and 3Be), where a higher percentage of SNH in the surrounding landscape supported a greater abundance of arthropods. Although there was no buffering effect observed for abundance patterns in CR (Fig 3Bc), higher percentages of SNH benefited arthropod abundance at least in sites with low levels of N deposition (N deposition < 1,000 mg N m⁻² yr⁻¹).

## Discussion

Earlier studies of the effects of N deposition on biodiversity mainly focused on plant diversity [24,26–28]. The few studies reporting effects of N deposition on local animal biodiversity either focused on a small selection of taxa [31,78–80], or studied responses observed under highly controlled N deposition rates [25]. Here, we demonstrated, for the first time, a negative effect of N deposition on arthropod diversity at the global scale across different land use types. We further highlighted the interactions between N deposition and landscape composition in influencing arthropod diversity, with particular focus on natural and semi-natural habitats.

### Variation in N deposition impacts across land-use types

Similar to earlier studies [7,38,81,82], we show that N deposition impacts on arthropod diversity are not acting uniformly across land-use types. Due to habitats dominated by primary and secondary natural vegetation commonly being located distant from major sources of N emissions and also historically experiencing low levels of N deposition [81], their arthropod diversity can be expected to show an enhanced sensitivity to N deposition when compared for example to assemblages in PF or on pastureland. As highlighted by previous research [30,79,81], the observed decline in animal diversity caused by N deposition is frequently linked to: (1) reduced plant diversity, which lowers food diversity; (2) increased plant biomass and density, which cool the microclimate and make habitat conditions less suitable for survival; and (3) changes in plant chemical composition, leading to reduced food quality. In addition, N deposition also interacts with management practices. For instance, higher intensity land use is per se associated both with biodiversity losses and high levels of N deposition and N application [83]. Responses to N deposition are further differentiated by differences in soil pH that drives specific plant responses to the depositions, which in turn affects animal species [30,84]. Therefore, the combined effects of these variables under different land-use types lead to a differential impact of N deposition on biodiversity across varying land-uses [85] (Fig 2).

The negative response of species richness to N deposition in CR could be partly related to long-term exposure to N deposition related to fertilization applications in combination with atmospheric deposition [86]. Higher N deposition also leads to soil N enrichment and nitrification [84,87], which may result in increased risk of pest outbreaks [38] that in turn require more intensive applications of pesticides that further affect biodiversity in anthropogenic habitats like CR, and this also occurs to PF and PA. As many species in landscapes with a high CR cover will have been filtered out [88], already, the species pool left at such sites may be highly stress-tolerant and pre-adapted to high levels of N exposure, resulting in limited diversity responses in such landscapes. In PA, the increased prevalence of N-demanding species at the expense of less competitive ones may lead to a reduction in food resources, thereby negatively impacting the diversity of arthropod communities [89]. In Central Europe, for example, plant species richness in grassland decreased by approximately 23% with increasing N deposition [89], a trend similar to our results for arthropod richness. N enrichment may also increase biomass accumulation in the vegetation that in turn changes the vegetation structure and can enhance fire risks during dry

spells [90], again potentially adversely affecting arthropod communities. It can be anticipated that changes in arthropod diversity within more natural habitats may primarily be driven by N deposition-induced shifts in plant community composition toward species less preferred by arthropods, or a homogenization and increased density of the vegetation that can shrink ecological niche space. In contrast, responses in human-dominated landscapes such as CR will more likely be influenced by habitat alteration caused by, or associated with enhanced N deposition. However, these interpretations remain hypothetical and require further empirical validation through specific case studies [29]. Nonetheless, it is clear that N deposition can not only have negative effects on biodiversity, but N addition may significantly increase arthropod biomass, with grassland eutrophication, for example, being linked to an increase in arthropod abundance [91,92]. Moreover, arthropod responses may vary depending on specific limiting nutrients in different environments [93].

### Importance of semi-natural and natural habitat

Consistent with our hypothesis, increasing the amount of SNH in the surrounding landscape partly buffers adverse effects of N deposition on arthropod biodiversity. These positive buffering effects might in the case of forests be related to their ability to directly alleviate the impacts of atmospheric N deposition on surrounding areas by intercepting N compounds in their complex canopies [21]. Meanwhile, some managed woodlands, for example, in agricultural landscapes, have not only a large leaf area, but create increased turbulence that helps dilute N deposition, thereby reducing the impact on the forest ground [43,94], but also on surrounding agricultural land, with these SNH acting as N sinks [21,95]. For example, in the UK and other parts of Europe, forest shelterbelts have been established in farmland to reduce the impacts of N deposition [96,97]. Similarly, compared to areas without them, areas with planted shrubs experience less biodiversity loss due to N deposition, as N is taken up and utilized more quickly by the plant community, reducing its direct accumulation in the soil and helping to mitigate biodiversity loss [98]. Furthermore, an increase in SNH generally increases the landscape heterogeneity, which benefits the conservation of a high species diversity at the landscape level by providing additional niches, resources, and potential refuge sites [44,99,100]. Therefore, preserving or establishing SNH across agricultural landscapes (dominated, e.g., by PF, PA or CR) offers a potential mitigation measure to counteract some of the negative effects of N deposition on biodiversity.

To further enhance the buffering effect, it is important to increase the proportion of SNH in the surrounding landscape. In line also with previous studies, some areas should be left 'for nature' [101–104] not only to conserve biodiversity and associated ecosystem services (e.g., biological pest control or pollination in landscapes used for food production), but also to buffer for the potential negative impacts of high N depositions. Meanwhile, although SNH in the surrounding landscape can buffer the negative impacts of N deposition, caution is needed not to extrapolate the results beyond the scope of the sampling. It should be noted that high N deposition in areas with a high proportion of SNH even has a positive effect (not just a buffering effect) (Fig 3). The result was in similar to a previous study, which reported the presence of shrubs and N deposition had an interactive effect on soil invertebrate diversity resulting in either an increase or a non-significant increase in invertebrate diversity when shrubs were planted [98]. However, considering our sampling sites rarely encompassed areas with both high proportions of SHN and high-level N deposition, these positive effects of SNH need to be validated with more field observations in the future.

### Implication for biodiversity conservation

Similar to previous studies on plant biodiversity [105,106], we found that deposition of reduced N, which is closely related to agricultural intensification, has a stronger negative impact on species diversity compared to oxidized N deposition. This likely linked to the aforementioned soil acidification due to deposition of these reduced forms of N [107]. Reduced (ammonium-dominated) N deposition will therefore alter soil microbial communities, plant secondary metabolites, and plant nutrient ratios more severely than oxidized N deposition, leading to plant damage and reduced plant quality, which in turn can affect arthropod food acquisition [29]. A further increase in reduced N deposition in the future [33,42], will likely

enhance direct losses of arthropod diversity. This highlights the importance of developing measures to reduce the negative impact of N deposition, and especially of reduced N deposition, on arthropod diversity [33,42]. It is therefore highly important to reduce NHx in particular, not least through changes in agricultural activities and fertilizer practices.

Targeted management aimed at enhancing plant and animal diversity could also include biomass removal through harvesting or grazing that mitigates N deposition effects by potentially directly removing some N, but also by increasing light availability, reducing vegetation height, and promoting heterogeneous microhabitats [61,108,109]. However, such disturbances may also cause direct mortality or temporarily reduce food resources [110]. In European grasslands and woodlands, liming is also commonly used to enhance N mineralization and reduce local N accumulation [111]. Furthermore, highly N-efficient agricultural practices and crops, and improved livestock management, may provide additional benefits [42]. Given the high N emissions from animal production, dietary shifts—such as replacing animal-based with plant-based or arthropod-based proteins—may be part of the solution. Our study highlights the need to better integrate the impacts of N deposition into biodiversity conservation targets—a factor that warrants greater attention [5], particularly regarding arthropods that are critical to ecosystem functioning and sustainable agriculture [79]. This systematic approach has gained heightened significance given the documented temporal escalation in N deposition rates [112], particularly in addressing biodiversity conservation challenges posed to arthropod populations.

### Limitations and future perspectives

It is worth noting that N deposition may impact animal communities at smaller spatial scales than we consider here [38]. Our large-scale analysis was limited by the coarse resolution of available N deposition data. Nevertheless, comparison with finer resolution data for one region (Europe) revealed a generally consistent trend in the effect of N deposition on arthropod species biodiversity (S2 Text and S9–S10 Tables, with a Pearson correlation coefficient between fine and coarse N deposition data of 0.85). Regardless of scales considered, N deposition was generally associated with biodiversity loss [113]. It is also noteworthy that regions experiencing the highest nitrogen deposition, such as parts of China and India, are severely under-represented in the PREDICTS database, which may lead to an underestimation of the negative impacts of extreme N deposition on biodiversity.

Additionally, our models' $R^2$ values were not very high (S3–S4 Tables), a trend reflected in earlier studies using the PREDICTS database [18,114]. Although previous studies have demonstrated greater variance explained by N deposition models than our model [27,89], the markedly lower explanatory power observed in our study may be attributable to two methodological distinctions: (1) the predominant focus on plant diversity (as opposed to arthropod diversity) in prior investigations, and (2) the comparatively restricted geographic scope of earlier research. The impact of N deposition on biodiversity was comparatively less significant than that of land use changes and agricultural intensification [11,12,50]. Nonetheless, even once controlling for effects linked to other variables known to affect biodiversity in our models, N deposition was still a significant explanatory variable. Therefore, while N deposition may not be the primary driver of arthropod diversity in agricultural and other systems, an effective N management aimed at reducing overall deposition rates is still an important component of global biodiversity conservation.

Future research is needed, under controlled experimental settings that reflect current and projected N deposition levels, to explore and test potential response mechanisms on arthropods, especially considering the cascading effects from changes in plant species [29]. Experimental sampling of arthropod communities is particularly needed in regions with high N deposition, such as China and India. On the other hand, it is also necessary to sample arthropods along gradients of N deposition—a fine-grained observational approach. Compared to high N additions in controlled experiments, such studies, which are closer to natural environmental conditions-will better support the conservation of species affected by atmospheric N deposition. In particular, comparative experiments between reduced and oxidized N deposition that separate the effects of these two forms will be more beneficial for implementing targeted biodiversity conservation measures.

## Conclusions

Our study highlights the necessity to address the detrimental effects of N deposition on biodiversity, as N deposition are still rising globally [42,112] and, according to our findings, are hence posing a significant and increasing threat to arthropod diversity. Failure to preserve and potentially expand SNH remnants across human-dominated landscapes may therefore result not only in significant arthropod diversity declines due to local N deposition, but will also impact diversity-associated ecosystem services like pollination and biological pest control that underpin a sustainable agricultural production. We therefore propose setting minimum SNH targets within production-dominant landscapes, while simultaneously promoting the ecological intensification of the production systems to safeguard biodiversity while securing sustainable yields even under the context of N deposition increases [49,115,116]. The impacts of N deposition we have demonstrated in our investigations also clearly highlight the importance of including reduction strategies for these deposition in agricultural production systems and environmental management approaches around the globe.

## Supporting information

**S1 Fig. Distribution of oxidized nitrogen deposition (NOy) and reduced nitrogen deposition (NHx) globally.** N deposition values were loge-transformed, with oxidized N deposition ranging from 0.7 mg N m$^{-2}$ yr$^{-1}$ to 3220.6 mg N m$^{-2}$ yr$^{-1}$, reduced nitrogen deposition ranging from 0.0 mg N m$^{-2}$ yr$^{-1}$ to 6576.8 mg N m$^{-2}$ yr$^{-1}$. The outline map is based on the Natural Earth (http://www.naturalearthdata.com/about/terms-of-use/), which is published under a CC-BY 4.0 license. The data underlying this figure can be found in link: https://thredds.met.no/thredds/catalog/data/EMEP/Articles_data/Schwede_etal_Ndep_2018/catalog.html.
(DOCX)

**S2 Fig. Distribution of N deposition sampling sites in Europe.** The outline map is based on the Natural Earth (http://www.naturalearthdata.com/about/terms-of-use/), which is published under a CC-BY 4.0 license. The data underlying this figure can be found in https://doi.org/10.6084/m9.figshare.29109170.
(DOCX)

**S3 Fig. The effects of the total N deposition interact with land use on Arthropods Chao estimated species richness.** Values represent the percentage difference compared with Primary vegetation with the 2.5th percentile value of total N deposition among sampled sites. The lines with different colors represent the median predicted value for each land-use type, with shaded areas representing the 95% confidence intervals. The results are predicted across 95% of the range of N deposition values for each land-use type. The number of sites were: Primary vegetation, $N_{site}$ = 1,094; Secondary vegetation, Nsite = 1,136; Cropland, $N_{site}$ = 1,285; Pasture, $N_{site}$ = 989; Plantation forest, $N_{site}$ = 277. The N deposition range used for plotting covers from 2.5% to 97.5% of sampled sites for each of the land-use types included in the model: 109.4–2326.3 mg N m$^{-2}$ yr$^{-1}$. The data underlying this figure can be found in https://doi.org/10.6084/m9.figshare.29109170.
(DOCX)

**S4 Fig. Model checks for the final species richness model obtained through backward stepwise selection for the study. A**, fitted versus residuals plot to check for constant variance across the range of fitted values. **B** is Normal QQ-plot to check for a normal distribution of residuals. **C** is distribution of *P*-values from sets of Moran's *I* test for spatial autocorrelation in the residuals for each study. Red line represents a *P*-value of 0.05. The left of red line represents the studies with significant spatial autocorrelation. **D** is observed values versus fitted values. The data underlying this figure can be found in https://doi.org/10.6084/m9.figshare.29109170.
(DOCX)

**S5 Fig. Model checks for the final total abundance model obtained through backward stepwise selection for the study. A**, fitted versus residuals plot to check for constant variance across the range of fitted values. **B** is Normal QQ-plot to check for a normal distribution of residuals. **C** is distribution of *P*-values from sets of Moran's *I* test for spatial autocorrelation in the residuals for each study. Red line represents a *P*-value of 0.05. The left of red line represents the studies with significant spatial autocorrelation. **D** is observed values versus fitted values. The data underlying this figure can be found in https://doi.org/10.6084/m9.figshare.29109170.
(DOCX)

**S1 Table. likelihood test result for species richness model.**
(DOCX)

**S2 Table. likelihood test result for total abundance model.**
(DOCX)

**S3 Table. The effects of the total nitrogen deposition on species richness of Arthropods.** Output mainly includes two parts: fixed part and random part. Fixed part includes estimates, 95% confidence intervals (CI) and *p*-values. Random part include: τ00 is the model variance explained by the random effects (SS represent studies, SSB represent blocks, SSBS represent sites within blocks), $\sigma^2$ is the residual variance, and the marginal and conditional $R^2$ values.
(DOCX)

**S4 Table. The effects of the total nitrogen deposition on total abundance of Arthropods.** Output mainly includes two parts: fixed part and random part. Fixed part includes estimates, 95% confidence intervals (CI) and *p*-values. Random part include: τ00 is the model variance explained by the random effects (SS represent studies, SSB represent blocks, SSBS represent sites within blocks), $\sigma^2$ is the residual variance, and the marginal and conditional $R^2$ values.
(DOCX)

**S5 Table. Effects of oxidized nitrogen deposition (NOy) on the species richness of Arthropods.** Output mainly includes two parts: fixed part and random part. Fixed part includes estimates, 95% confidence intervals (CI) and *p*-values. Random part include: τ00 is the model variance explained by the random effects (SS represent studies, SSB represent blocks, SSBS represent sites within blocks), $\sigma^2$ is the residual variance, and the marginal and conditional $R^2$ values.
(DOCX)

**S6 Table. Effects of reduced nitrogen deposition (NHx) on the species richness of Arthropods.** Output mainly includes two parts: fixed part and random part. Fixed part includes estimates, 95% confidence intervals (CI) and *p*-values. Random part include: τ00 is the model variance explained by the random effects (SS represent studies, SSB represent blocks, SSBS represent sites within blocks), $\sigma^2$ is the residual variance, and the marginal and conditional $R^2$ values.
(DOCX)

**S7 Table. Effects of oxidized nitrogen deposition (NOy) on the total abundance of Arthropods.** Output mainly includes two parts: fixed part and random part. Fixed part includes estimates, 95% confidence intervals (CI) and *p*-values. Random part include: τ00 is the model variance explained by the random effects (SS represent studies, SSB represent blocks, SSBS represent sites within blocks), $\sigma^2$ is the residual variance, and the marginal and conditional $R^2$ values.
(DOCX)

**S8 Table. Effects of reduced nitrogen deposition (NHx) on the total abundance of Arthropods.** Output mainly includes two parts: fixed part and random part. Fixed part includes estimates, 95% confidence intervals (CI) and *p*-values. Random part include: τ00 is the model variance explained by the random effects (SS represent studies, SSB represent blocks, SSBS represent sites within blocks), $\sigma^2$ is the residual variance, and the marginal and conditional $R^2$ values.
(DOCX)

**S9 Table. The effects of nitrogen deposition on the species richness of Arthropods in Europe with a finer resolution dataset (0.1°×0.1°).** Output mainly includes two parts: fixed part and random part. Fixed part includes estimates, 95% confidence intervals (CI) and $p$-values. Random part includes: τ00 for the model variance explained by the random effects (SS represent studies, SSB represent blocks, SSBS represent sites within blocks), $\sigma^2$ for the residual variance, and the marginal and conditional $R^2$ values. The oxnrdnRS represent the nitrogen deposition data derived from EMEP model. (DOCX)

**S10 Table. The effects of nitrogen deposition on the total abundance of Arthropods in Europe with a finer resolution dataset (0.1°×0.1°).** Output mainly includes two parts: fixed part and random part. Fixed part includes estimates, 95% confidence intervals (CI) and $p$-values. Random part includes: τ00 for the model variance explained by the random effects (SS represent studies, SSB represent blocks, SSBS represent sites within blocks), $\sigma^2$ for the residual variance, and the marginal and conditional $R^2$ values. The oxnrdnRS represent the nitrogen deposition data derived from EMEP model. (DOCX)

**S11 Table. The effects of the total nitrogen deposition on Chao estimated species richness of Arthropods.** Output mainly includes two parts: fixed part and random part. Fixed part includes estimates, 95% confidence intervals (CI) and $p$-values. Random part includes: τ00 for the model variance explained by the random effects (SS represent studies, SSB represent blocks, SSBS represent sites within blocks), $\sigma^2$ for the residual variance, and the marginal and conditional $R^2$ values. (DOCX)

**S12 Table. Variance inflation factors for the explanatory variables in richness model.** (DOCX)

**S13 Table. Variance inflation factors for the explanatory variables in abundance model.** (DOCX)

**S1 Text. Model structure and selection.** (DOCX)

**S2 Text. Test of finer N deposition data with 0.1° resolution.** (DOCX)

**S3 Text. The effects of the total N deposition on species richness of Arthropods when Chao estimated species richness index was used.** (DOCX)

## Acknowledgments

We thank David Makowski from University Paris-Saclay for his assistance with the statistical analysis in the paper.

## Author contributions

**Conceptualization:** Shunxiang Fan, Yunhui Liu.

**Funding acquisition:** Yunhui Liu.

**Methodology:** Shunxiang Fan.

**Supervision:** Tim Newbold, Yunhui Liu.

**Writing – original draft:** Shunxiang Fan.

**Writing – review & editing:** Shunxiang Fan, Tim Newbold, Jan C. Axmacher, Charlotte L. Outhwaite, Yi Zou, Zhenrong Yu, Yunhui Liu.

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
