## [Editor Report · Decision Letter 0]

Dear Dr Liu,

Thank you for submitting your manuscript entitled "Landscape composition mediates global losses in arthropod diversity caused by atmospheric nitrogen deposition" for consideration as a Research Article by PLOS Biology.

Your manuscript has now been evaluated by the PLOS Biology editorial staff, as well as by an academic editor with relevant expertise, and I'm writing to let you know that we would like to send your submission out for external peer review.

Once your full submission is complete, your paper will undergo a series of checks in preparation for peer review. After your manuscript has passed the checks it will be sent out for review. To provide the metadata for your submission, please Login to Editorial Manager (https://www.editorialmanager.com/pbiology) within two working days, i.e. by Mar 25 2025 11:59PM.

Kind regards,

Roli Roberts

Roland Roberts, PhD

Senior Editor

PLOS Biology

rroberts@plos.org

---

## [Decision Letter · Decision Letter 1]

Dear Dr Liu,

Thank you for your patience while your manuscript "Landscape composition mediates global losses in arthropod diversity caused by atmospheric nitrogen deposition" went through peer-review at PLOS Biology. Your manuscript has now been evaluated by the PLOS Biology editors, an Academic Editor with relevant expertise, and by several independent reviewers.

You'll see that reviewer #1 is positive, but flags two apparent inconsistencies which will require further discussion. Reviewer #2 is also positive but has some more substantial concerns; these include the framing of the work (point 1) and the Discussion (points 4 and 5), but also may involve further analyses (points 2 and 3). S/he has a long list of textual issues. Reviewer #3 is very positive, and most of his/her comments relate to the structure and content of the Intro and Discussion (there are one or two potential analyses).

In light of the reviews, which you will find at the end of this email, we are pleased to offer you the opportunity to address the comments from the reviewers in a revision that we anticipate should not take you very long. We will then assess your revised manuscript and your response to the reviewers' comments with our Academic Editor aiming to avoid further rounds of peer-review, although we might need to consult with the reviewers, depending on the nature of the revisions.

**IMPORTANT - SUBMITTING YOUR REVISION**

*Resubmission Checklist*

*Published Peer Review*

*PLOS Data Policy*

*Blot and Gel Data Policy*

Sincerely,

Roli Roberts

Roland Roberts, PhD

Senior Editor

PLOS Biology

rroberts@plos.org

REVIEWERS' COMMENTS:

Reviewer #1:

Dear Authors, dear Editor,

In this study, the authors inferred the effects of nitrogen (N) deposition on arthropod diversity by integrating biodiversity data from the PREDICTS database with global N deposition and land cover information. The findings reveal that N deposition negatively impacts arthropod biodiversity, with significant reductions in species richness and abundance, particularly in less anthropogenically modified landscapes.

Thank you for the opportunity to review this interesting study. In my opinion, it is a very well conducted study with clearly described methods and results. The findings are very intresting and support previous studies that demonstrate negative effects of N deposition not only on plants but also on other species groups. I only have few comments:

Main comments

- Based on the presented results, the authors conclude that the negative effect of N deposition can be buffered by a large proportion of semi-natural and natural habitats (SNH) in the surrounding area (see results from line 396). This is an important finding. However, when examining Fig. 3, it appears that high N deposition in areas with a high proportion of SNH even has a positive effect (not just a buffering effect). It is unclear to me whether these are actual effects or could be artefacts. Nevertheless, this is a point that should be discussed in more depth.

- The authors find that the effect of nitrogen deposition is most pronounced in habitats with the least anthropogenic modifications (l. 26-27), and at the same time, an increase in the proportion of natural habitat reduces the negative effect of nitrogen deposition (l. 27-29). To me, this sounded like a contradiction, particularly when first reading the abstract. It should be emphasised more strongly that the former refers to the habitat at the study area, while the latter pertains to the proportion of natural habitat in the surrounding area.

Minor comments

- l. 25: slightly misleading as the N-deposition was not measured in the field at each site.

- l. 28: Add "increases in SNH proportion IN THE SURROUNDINGS reduced ..."

- l.47: Please explain why global-scale effects of N deposition require greater attention.

- l.165-167: The abundance was corrected for the sampling effort. However, such a correction does not appear to have been made for the species richness.

- l. 200: Should the N-deposition model be included as a factor in the statistical model?

- l. 201: Remove Tan et al. in the brackets

- l. 243-244: Check wording of the sentence.

- l. 340: The link does not seem to be working.

- l. 578: Please add a reference that N deposition is still rising globally.

Reviewer #2:

This article mainly address the how the landscape composition, especially the proportion of natural and semi-natural habitats (SNH), mediate the negative effects of nitrogen deposition on arthropod diversity at a global scale, and the main findings are that N deposition has negative effects on arthropod diversity, and such effect differ across land use types. The most impressive and interesting finding is that the presence and large proportion of SNH in the landscape could offset the negative effects of N deposition on arthropod diversity, which in my view is the novelty of the study. Based on such buffering effect, the authors propose the need to maintain a minimum proportion of natural and semi-natural habitats especially in agricultural landscape, from the perspective of biological conservation and thus the sustainability of ecosystem services. In my view, the strength of the paper lies in its global scale examination and its core finding of buffering effects of SNH for biodiversity loss due to N deposition. However, I still have some major concerns and many minor ones as listed below:

1. The differentiation and the comparison between the effects of reduced N and oxidized N seems a finer question compared to the major theme of the paper. Such differentiation and the comparison may more or less divert readers' attention from your most interesting and novel thematic finding. Especially, in the first scientific question, it come abruptly, as it seems that no backdrop for it was previously set.

2. I worry about the match between arthropod assemblage data and land use data, as the former spans 1984-2013, while the latter is from a single year, 2005. After all, the arthropod assemblages would change very quickly following land use conversion, which also happen very abruptly due to human interference.

3. Since you have used data of climate and soil for each site on a global scale, and these data must span a large range, so I am wondering why you did not analyze how the N deposition effect and/or SNH buffering effects were influenced by the climate/soil factors. For example, it is very probably that such effects may differ between dry and wet regions.

4. The Discussion section is not so well organized, that is, the contents are arranged not in a good logic, and there are quite some contents diverting from the main theme (such as line 556-558). As a consequence, the entire Discussion is not very cohesive. For example, line 432-444, this is about the comparison between the effects of reduced N and oxidized N, which is a finer question compared to the major theme of the paper, but such content was placed in the very beginning of specific discussion. If you maintain this content (please also see my first concern), please postpone it. Moreover, the subsequent part of this paragraph (line 444-459) is about the implication of your results for biodiversity conservation, which should be postponed to the end of Discussion or to a separate section named "Implications", together with other implications such as "conserving or expanding SNH" (line 531-541).

5. You devoted only one paragraph (line 503-521) to discuss the mechanisms of SNH buffering effects, which is in my view far from adequate, because this is your core and novel finding. I suggest to expand this part while briefing other parts especially those that seems too hypothetical or a bit distant from your main theme.

Other minor concerns:

1. I suggest the title be changed into "Landscape composition mediates the negative effects of nitrogen deposition on arthropod diveristy"

2. Line 25, delete "observed"

3. Line 26, for → across

4. Line 42, delete "rates"

5. Line 50, delete ", too"

6. Line 52, of arthropods, the world's most species-rich taxa, are therefore……

7. Line 62, at → for

8. Line 63, exist → coexist

9. Line 67, delete "in response to increased N deposition"

10. Line 68, insert "On the other hand"

11. Line 72, to → for

12. Line 75, delete "that", insert "and" before play

13. Line 80, put forward the comma after "and" to before it

14. Line 94-95, please delete this sentence, as this is another issue beyond this paper

15. Line 104, Consequence → Therefore

16. Line 107, delete "while also"

17. Line 111, insert "and" before challenging

18. Line 120, semi-natural/natural habitats, proportion of cropland → SNH and cropland

19. Line 123, "Differentiated by land-use type" → "Across land-use types"

20. Line 126, extent → proportion

21. Line 130, delete "furthermore"

22. Line 130, I don't understand this sentence, "most negative effect" is in comparison with the effects of oxidized N? what do you mean by "not least due to ……"

23. Line 131, the second hypothesis can be changed into "SNH can buffer the negative effect of N deposition on biodiversity"?

24. Line 153, delete "measures of"

25. Line 159, accounted for → encompassed

26. Line 166-167, it is not necessarily the linear relationship, but very probably the asymptote, like the species number vs. area curve.

27. Line 192, from → of

28. Line 202, value should be in singular form

29. Line 207, onto → into

30. Line 213, the semicolon should be comma

31. Line 220-221, can be changed into "we additionally compared the responses of arthropod diversity to the two types of N deposition separately"?

32. Line 226, better combine these two sentences, by insert ", and found that"

33. Line 232, "biodiversity" should be "land use"?

34. Line 235, MAT and MAP data of only one year 2010? Does it match with the land use data of 2005? And with arthropod data of 1984-2013?

35. Line 241, usage → use

36. Line 243, noted worth → noteworthy

37. Line 249, pH

38. Line 253, which four variables?

39. Line 263, based on their relationships

40. Line 270, percentage of

41. Line 276, oxidized and reduced

42. Line 279, to convert these → as to be converted

43. Line 285, with → into

44. Line 300, the interaction effect → its interactive effect

45. Line 366, Table S3 should be deleted?

46. Line 367, "ranked the second"? in terms of what? Is it possible for both the secondary vegetation and pasture to rank the second? By the way, please insert "of arthropod diversity" after "Negative responses"

47. Line 368, delete "diversity"

48. Line 370-371, the phase "as well as……" is wrongly used, in terms of syntax. How about the primary and secondary vegetation? It is not possible for them to suffer greatest loss as aforementioned pasture!

49. Line 373, compared to that in sites of primary vegetation

50. Line 376, more strong → stronger

51. Line 397-410, in this para, is the first sentence contradictory with the last sentence with respect to the buffering effect of SNH? Moreover, all the words "biodiversity" refer to diversity of arthropods? If so, please specify it as arthropod diversity. There do exist same problems in some other places of the MS.

52. Line 406-407, this is not a complete sentence!

53. Line 423, delete "has"

54. Line 426, delete "experiments data"

55. Line 429, furthermore → further

56. Line 436, uniquely liked to → due to

57. Line 438, delete "subsequently"

58. Lien 441, could → might

59. Line 442-43, delete "and" and "has",

60. Line 447, species richness! But do you mean plant species here, or arthropod species? Please specify!

61. Line 447-448, by grazing that can offset light limitation

62. Line 454, delete "extensive of"

63. Line 457-459, what are such measures?

64. Line 465-466, can be changed into: ", their biodiversity may show higher sensitivity to N deposition than plantation forest or pastures"

65. Line 468, "reduced edibility of plant resources"? why? Don't plants of higher N content have better edibility

66. Line 470, semicolon → ", for instance,"

67. "per se" should be italicized

68. Line 471, high levels of exposure to N deposition and N application

69. Line 480, delete "direct"

70. Line 483, "but also plantation forests and pasture" → "and this also occurs to plantation forests and pasture"

71. Line 490, plant or arthropod species richness?

72. Line 493 could → can

73. Line 498, "providing a reduction in" → "that can shrink"

74. Line 503, delete "the"

75. Line 505, delete "coverage"

76. Line 506, "reduce" → "alleviate the"

77. Line 507, "removing" → "intercepting";

78. Line 513, "from" → "due to"

79. Line 514, "as N will not accumulate in the soil", why not?

80. Line 516, "contributes positively towards" → benefit

81. Line 520, "address" → "counteract"

82. Line 521, "N deposition exerts on biodiversity" → "of N deposition on biodiversity"

83. Line 528, "between" → "of"

84. Line 529, "and" → "including"

85. Line 542, delete "furthermore"

86. Line 548, regardless of scales considered

87. Line 556-558, delete this sentence

88. Line 559-561, The prospect for future research should be focused on those especially at relevant scale as the present study, or with its combining with manipulative experiments to uncover the underlying mechanisms.

89. Line 569, being → was

90. Line 571, "accounting for" → "identifying"

91. Line 587, "in view of" → "under the context of"

Reviewer #3:

[identifies himself as David Wagner; IMPORTANT: please see attached file]

This is a much-needed look at the impacts of nitrogen inputs on terrestrial arthropod diversity. Figures are fine. The rigor of the study is reflected in the four supplementary documents. The results make sense, are novel, important, consistent with reports from other studies, and are of global relevance. I suspect that the paper will be highly cited.

I am not sufficiently knowledgeable about the statistical methods to evaluate aspects of the study. Pleased to see that Fan et al. also took mean annual temperature, mean annual precipitation, and soil PH into account (and used variance inflation factors to check for collinearity). Likewise, was pleased that the team analyzed organic oxides separately from reduced nitrogen cmpds.

Most of my comments and suggestions are added to a pdf of the manuscript which I am returning.

The Discussion struck me as a bit long and scattershot. Consider shortening and perhaps add a few subsections or header words that organize key points so that readers can find relevant content easily given that the Discussion touches on many subjects.

The Introduction and or Discussion need to include text that is broad in scope about how N inputs change the basic ecology of ecosystems: primary productivity, species composition, vegetation structure, litter load, duff characteristics, fire, and perhaps most fundamentally in terrestrial ecosystems, the soil properties (a possible reference https://www.sciencedirect.com/science/article/pii/S0038071722003443), far beyond just pH. As such nitrogen input can and will foundationally trigger changes in the composition of a tract's microfauna, flora, and fauna. And from there tell reader that the Discussion can/will only treat a few examples of such, e.g., some of the effects that have been well studied/documented. This approach would help fix the scattershot feeling I got from reading the Discussion.

Related to the above. Be a bit more even-handed about discussing the consequences of the changes. Not all changes are inherently bad--ecological changes produce losers and winner. For example, it is not only the pests that benefit from N input as implied by the current wording. Likely reckon that at least few imperiled taxa might also do better in a situation with more nitrogen.

The Discussion would benefit from pointing out the there are many terrestrial and freshwater systems globally that are especially N-limited, where atmospheric and dry deposition from the use of the burning of fossil fuels represents an extremely threat to the autochthonous biodiversity. The organisms are specially adapted to survive in such situations, and many are easily displaced once the soils are changed by N inputs…even in wildlands, far away from agriculture and anthropogenic point sources.

It is important to emphasize the need for more experimental studies. This could be a separate paragraph; not be handled as just a sentence or two embedded in a larger paragraph.

Small matters:

* The land use types are capitalized in some places but not others. Edit for uniformity. In this manuscript capitalization might be best.

* Text sometimes equates species richness with biodiversity. That's a little problematic and the authors would be well served by just sticking to using richness. If they are to regard them as synonyms, they should be explicit that they plan to do such somewhere.

Lines 40-41: atmospheric deposition as the inclusive term is a misnomer. Come up with another term for pollution that includes agricultural inputs. Maybe anthropogenic inputs, anthropogenic augmentation, or any other word set that less problematically includes atmospheric deposition + fertilizer application(s).

Lines 64 to 68: might consider citing Wagner, D. L. (2020) [Insect declines in the Anthropocene. Ann. Rev. Entomol. 65: 457-480] here/elsewhere as N pollution received its own subsection in the review. (Text reproduced below.) (Being a biosystematist with training in botany and a rare species biologist, nitrogen pollution scares the pudding out me, and makes me appreciate the gravity of this study.)

Nitrification

The release of atmospheric nitrites from the burning of fossil fuels has the potential to change global ecology and ignite extinction cascades (121, 134). Nitrogen is often a limiting nutrient in terrestrial, freshwater, and marine systems. Many of the world's most speciose and specialized plant communities occur in nitrogen-poor soils (158). In a now-classic study, the generous addition of nitrogen to a pasture at the Rothamsted Experimental Station (in the United Kingdom) resulted in a 10-fold loss of plant richness over the course of the ensuing century (77).

The wet or dry deposition of atmospheric nitrogen to previously nitrogen-limited communities (e.g., bogs, heathlands, sand-based communities, pine barrens, oligotrophic grasslands, and many freshwater ecosystems) will change their fundamental ecology by altering edaphic conditions, nutrient cycling, species interactions and composition, and more. Butterfly declines in northern Europe have been linked to nitrogen deposition, particularly that associated with oligotrophic grasslands (56a, 57, 114).

Soil chemistry is a primary determinant of plant distributions—nitrogen deposition from fossil fuel consumption has the potential to trigger changes in plant communities worldwide. As plant abundance and species composition change, so will the associated entomofaunas. Approximately half of all described insects are plant feeders, and approximately 90% of these are thought to be dietarily specialized herbivores that depend upon just a small suite of closely related species (37, 127, 148). The potential consequences of atmospheric nitrogen deposition are grave and worthy of greater attention.

Line 96: Agriculture (agricultural intensification) is identified as one of the top three drivers of insects declines in Wagner (2020), Wagner et al. (2021), and Raven and Wagner (2021). None of these need be cited…just sayin'.

Lines 219-220: The effects of the two forms of N (input) do differ—not may differ.

Lines 243-247: In parts of the USA more fertilizer is deployed on lawns than in croplands! And still further inputs come from impervious surfaces in towns and cities. Not sure these would affect the results as most of the study sites would be outside such areas. But it might be a good idea to analyze the sites within vs away from suburbia and cities, separately, at some point. I am not suggesting that this be done at this time, unless authors were interested in doing so.

Line 429: There are over 10 million species of insects--why is that a small selection of taxa? Also, as noted above, the issue of nitrogen as a driver is addressed in more detail in Wagner (2020). Wagner et al. (2021) might be deleted here--it would be more relevant to line 79.

Lines 438-439: "subsequently potentially disrupting arthropod trophic networks." This is almost a truism. Every scenario, changes foodwebs: any inputs, withdrawals, time, weather, etc. Somehow this implies to me that these systems are in some stable state that is being disrupted. Everything changes foodwebs; they are inherently dynamic, especially in early successional systems. Consider rephrasing to put something besides the obvious into the statement.

Lines 444-459: a bit wordy and scattered--consider tightening this text up...

Line 467-468: structural changes can increase or decrease insect diversity. The text is too non-specific. Rework text. Tighten up.

Lines 472-472: As written, the sentence is a truism. Of course, soil pH influences biodiversity… tightened up.

Lines 508-511: Mineralization is not mentioned in the manuscript. This is a exceedingly important way a grassland, woodland, or forest can remove organic nitrogen from the landscape, and needs to be mentioned somewhere.

Lines 523-524: Straw man as written. Agencies/NGO's/governments don't focus on just one thing. Conservation biologists understand these matters to a greater degree than implied here, and that insecticides, herbicides, fungicides, invasive species, other forms of pollution, etc. are all in play.

Line 524: Estimation of rates is another matter entirely and does not (so much) depend on one's beliefs. Perhaps rewrite sentence to express that you feel this important stressor is underappreciated and understudied.

Question: Might there be latitudinal gradients in fertilizer use (greater use in temperate areas) and in species richness. To what extent are these intertangled or not in this study?

As noted above, the Discussion struck me as a tad long, places where adjacent sentences could be combined, authors could dial back on indefinite pronouns, and reword sentences that read as truisms. Polish up a bit to deliver crisp, tight takeaways that readers can digest.

The study is a "satellite's view" of nitrogen deposition and its impacts on insect diversity and abundance. The authors make a pitch for the need of experimental work, which is much needed. But I wonder if there shouldn't also be pitch for studies that fit in between these two scales. For example, local studies that examine insect diversity across nitrogen gradients—a finer-grained approach. Might the authors in their Discussion/Conclusion make a plea for such.

In sum, I view this an especially worth study of great relevance to the biodiversity crisis and the matter of insect decline, and I am much in favor of it going forward. Nearly, all my suggestions are merely that and need not be embraced.

---

## [Editor Report · Decision Letter 2]

Dear Dr Liu,

Thank you for your patience while we considered your revised manuscript "Landscape composition mediates negative effects of nitrogen deposition on arthropod diversity" for publication as a Short Reports at PLOS Biology. This revised version of your manuscript has been evaluated by the PLOS Biology editors and the Academic Editor.

Based on our Academic Editor's assessment of your revision, we are likely to accept this manuscript for publication, provided you satisfactorily address the following queries from the AE, data and other policy-related requests.

IMPORTANT - Please attend to the following:

a) Please change your Title to something that is more accessible, declarative, and more closely aligned with the Abstract. We suggest: "Arthropod biodiversity loss from nitrogen deposition is buffered by natual and semi-natural habitats"

b) Please address the remaining requests from the Academic Editor (see the foot of this email). Specifically, the AE wants you to incorporate two elements of your rebuttal into the main paper. Also the AE and I were both puzzled as to why Figures S4C and S5C differ between the R1 and R2 versions of the manuscript; please explain.

c) Please address my Data Policy requests below; specifically, we need you to supply the numerical values underlying Figs 1, 2AB, 3ABCDEABCDE, S1, S2, S3, S4ABCD, 5ABCD, either as a supplementary data file or as a permanent DOI’d deposition. I note that you already have an associated Figshare deposition (DOI:10.6084/m9.figshare.29109170), but this is currently not accessible. Please make it accessible or send me a reviewer link so that I can check its compliance with our policy.

d) Please cite the location of the data clearly in all relevant main and supplementary Figure legends, e.g. “The data underlying this Figure can be found in S1 Data” or “The data underlying this Figure can be found in DOI:10.6084/m9.figshare.29109170"

e) I note that you mention two of the reviewers (“David Wagner and two anonymous reviewers”) in the Acknowledgements. While we appreciate the sentiment, this is against PLOS policy, so please could you remove this?

f) Please make any custom code available, either as a supplementary file or as part of your Figshare data deposition.

We expect to receive your revised manuscript within two weeks.

*Published Peer Review History*

*Press*

Sincerely,

Roli Roberts

Roland Roberts, PhD

Senior Editor

rroberts@plos.org

PLOS Biology

DATA POLICY:

Regardless of the method selected, please ensure that you provide the individual numerical values that underlie the summary data displayed in the following figure panels as they are essential for readers to assess your analysis and to reproduce it: Figs 1, 2AB, 3ABCDEABCDE, S1, S2, S3, S4ABCD, 5ABCD. NOTE: the numerical data provided should include all replicates AND the way in which the plotted mean and errors were derived (it should not present only the mean/average values).

CODE POLICY

COMMENTS FROM THE ACADEMIC EDITOR:

I have looked over the authors' responses. I agree that they have been quite serious in their responses to the host of requests from the reviewers.

However, there are a few areas where they could be more upfront in the main text, particularly where they have been asked to defend their use of the data and consider a re-analysis but have not. For the former, they have repeatedly cited the Nature paper, that a few of their co-authors are on, as a reason for the temporal mismatch in terms of the data (R1 1.200). I would like them to include the rationale in the main text. For the latter, the authors often raised the potential issue of (un)interpretability if the model has too many higher-order interactions and is too complex as a reason for not re-analyzing (R2 2 and 3). They would have to directly mention this in the main text as well. Lastly, I am as puzzled as you are about Figure S4C and S5C so they would need to explain why the two figures were replaced.

---

## [Editor Report · Decision Letter 3]

Dear Dr Liu,

Thank you for the submission of your revised Short Report "Arthropod biodiversity loss from nitrogen deposition is buffered by natural and semi-natural habitats" for publication in PLOS Biology. On behalf of my colleagues and the Academic Editor, Tien Ming Lee, I'm pleased to say that we can in principle accept your manuscript for publication, provided you address any remaining formatting and reporting issues. These will be detailed in an email you should receive within 2-3 business days from our colleagues in the journal operations team; no action is required from you until then. Please note that we will not be able to formally accept your manuscript and schedule it for publication until you have completed any requested changes.

Sincerely, 

Roli Roberts

Senior Editor

PLOS Biology

rroberts@plos.org